# Maximum Respiration Rates in Hyporheic Zone Sediments are Primarily Constrained by Organic Carbon Concentration and Secondarily by Organic Matter Chemistry

James C. Stegen[1], Vanessa A. Garayburu-Caruso[1], Robert E. Danczak[1], Amy E. Goldman[2], Lupita Renteria[1], Joshua M. Torgeson[2], and Jacqueline Wells[2]

[1]Earth and Biological Sciences Directorate, Pacific Northwest National Laboratory, Richland, WA, U.S.A

[2]Energy and Environment Directorate, Pacific Northwest National Laboratory, Richland, WA, U.S.A

*Correspondence to*: James C. Stegen (James.Stegen@pnnl.gov)

**Abstract.**

River corridors are fundamental components of the Earth system, and their biogeochemistry can be heavily influenced by processes in subsurface zones immediately below the riverbed, referred to as the hyporheic zone. Within the hyporheic zone, organic matter (OM) fuels microbial respiration, and OM chemistry heavily influences aerobic and anaerobic biogeochemical processes. The link between OM chemistry and respiration has been hypothesized to be mediated by OM molecular diversity, whereby respiration is predicted to decrease with increasing diversity. Here we test the specific prediction that aerobic respiration rates will decrease with increases in the number of unique organic molecules (i.e., OM molecular richness, as a measure of diversity). We use publicly available data across the United States from crowdsourced samples taken by the Worldwide Hydrobiogeochemical Observation Network for Dynamic River Systems (WHONDRS) consortium. Our continental-scale analyses rejected the hypothesis of a direct limitation of respiration by OM molecular richness. In turn, we found that organic carbon (OC) concentration imposes a primary constraint over hyporheic zone respiration, with additional potential influences of OM richness. We specifically observed respiration rates to decrease nonlinearly with the ratio of OM richness to OC concentration. This relationship took the form of a constraint space with respiration rates in most systems falling below the constraint boundary. A similar, but slightly weaker, constraint boundary was observed when relating respiration rate to the inverse of OC concentration. These results indicate that maximum respiration rates may be governed primarily by OC concentration, with secondary influences from OM richness. Our results also show that other variables often suppress respiration rates below the maximum associated with the richness-to-concentration ratio. An important focus of future research will identify physical (e.g., sediment grain size), chemical (e.g., nutrient concentrations), and/or biological (e.g., microbial biomass) factors that suppress hyporheic zone respiration below the constraint boundaries observed here.

## 1 Introduction

River corridors are key components of the Earth system that connect terrestrial landscapes to the ocean through the transport and transformation of organic matter (OM) and nutrients (Harvey and Gooseff, 2015; Schlünz and

Schneider, 2000; Schlesinger and Melack, 1981). In addition, river corridors have strong connections to the atmosphere in terms of significant emissions of greenhouse gasses such as $CO_2$, contributing $\sim$3.9 Pg $CO_2$-C yr$^{-1}$ to the atmosphere (Raymond et al., 2013; Drake et al., 2018). Within river corridors the hyporheic zone (Orghidan, 2010) can have a dominant influence over net metabolism and biogeochemical transformations (Boulton et al., 1998; Naegeli and Uehlinger, 1997; Krause et al., 2011). Here we define the hyporheic zone as shallow subsurface sediments through which surface water enters, moves through and at some point returns to the main channel. These zones are considered biogeochemical "hotspots" and can be responsible for 3-96% of the total stream metabolism (Jones, 1995; Fuss and Smock, 1996; Naegeli and Uehlinger, 1997; Kaplan and Newbold, 2000; Battin et al., 2003; Ward et al., 2018). In turn, hyporheic zones can contribute significantly to the overall $CO_2$ emissions from inland waters (Burrows et al., 2017; Newcomer et al., 2018; Comer-Warner et al., 2019; Son et al., 2022). Recent work has found that detailed properties of OM chemistry can significantly influence respiration rates in hyporheic zone sediments (Stegen et al., 2018; Garayburu-Caruso et al., 2020a; Sengupta et al., 2020; Graham et al., 2018, 2017; Song et al., 2020). These observations demonstrate a need to deepen understanding of the relationships between hyporheic zone biogeochemistry (e.g., respiration rates) and OM chemistry.

A conceptual hypothesis was recently developed that may provide new insight into the connections between OM chemistry and biogeochemical rates. More specifically, Lehmann et al. (2020) hypothesize that OM can be protected from degradation (in part) by high levels of molecular diversity. Biogeochemical rates that depend on OM oxidation (e.g., aerobic respiration) may therefore be suppressed with increases in the number of unique organic molecules (referred to here as OM molecular richness). The concept is that high levels of OM molecular richness lead to low returns-on-investment, relative to the energy invested in building and maintaining the molecular machinery needed to metabolize any given type of organic molecule. The consequence is low respiration rates. The underlying mechanism has been proposed to help protect OM in some ecosystems such as deep sea (Arrieta et al., 2015) and river corridor (Stegen et al., 2018) environments.

The hypothesis of lower biogeochemical rates with higher OM molecular richness has not been evaluated in hyporheic zone sediments despite the established connection between OM chemistry and hyporheic zone respiration rates. We posit that higher levels of hydrologic connectivity in hyporheic zones relative to unsaturated systems (e.g., soil) may diminish influences of spatial isolation such as an OM stabilization mechanism (Schmidt et al., 2011), potentially leading to particularly strong relationships between respiration rates and OM chemistry. In turn, it is plausible that the hyporheic zone is an ecosystem in which we may find support for the hypothesized negative relationship between respiration rates and OM molecular richness. Here we test this hypothesis at the continental scale using publicly available data from the Worldwide Hydrobiogeochemical Observation Network for Dynamic River Systems (WHONDRS) consortium (Stegen and Goldman, 2018; Garayburu-Caruso et al., 2020b; Toyoda et al., 2020; Goldman et al., 2020).

## 2 Methods

*Sample collection and data generation*

During the summer of 2019, the WHONDRS consortium carried out a multi-continent river corridor study to evaluate interactions between metabolomes, microbial metabolism, biogeochemical function, and ecosystem features. Garayburu-Caruso et al. (2020b) describe details on metadata, sample collection, analysis, and processing of ultrahigh resolution mass spectrometry data. Briefly, during late July and August 2019 sediment samples were collected across multiple continents, but the current study focuses on samples collected in the contiguous United States (ConUS) (Fig. 1). Sampled locations spanned a broad range of environmental conditions; for example, stream order ranged from 1st to 8th, land cover composition varied with upstream forest cover ranging from 0-97 % and urban cover ranging from 0-28%, and physical settings were from relatively steep headwater streams to large lowland rivers. At each site, shallow sediments (~1-3 cm depth) were collected at three separate depositional zones. The zones were ~ 10 m away from each other and were labeled as upstream, midstream, and downstream. Samples were shipped to the Pacific Northwest National Laboratory (PNNL) campus in Richland, WA (USA) on ice within 24 hours of collection. We conceptualize these sediments as part of the hyporheic zone as we make the assumption that the supply and exchange of nutrients and OM from the stream influences the biogeochemical processes experienced by the sediments. In turn, we assume that all samples came from shallow (~1-3 cm depth) hyporheic zone sediments through which surface water enters, moves through, and at some point returns to the main channel.

In the laboratory, sediments were sieved with a 2 mm sieve, and subsampled into 50 mL conical tubes (Genesee Scientific Olympus™ Plastics) to separate Field and Incubation aliquots. Note that in the methods provided by Garayburu-Caruso et al. (2020b) there is an error in the description of the sediment preservation prior to mass spectrometry analysis. Corrected preservation methods are described immediately below. Sediments from the Field aliquot were flash frozen in liquid nitrogen immediately after sieving to maintain the sediment characteristics observed in the field and stored at -80°C until analysis. The Incubation aliquots were not flash frozen immediately; instead they were kept in the dark inside an environmental chamber at 21°C along with other sediments to be used for respiration measurements (see below) so that the two sets of sediment samples experienced the same conditions leading up to the use of the sediment for respiration estimation. The next morning, Incubation aliquots were retrieved from the environmental chamber, flash frozen in liquid nitrogen, and stored at -80°C until analysis. In our analyses we used the "Field" sediments to study water-extractable organic carbon concentration and OM chemistry prior to the respiration incubation. We used the "Incubation" sediments as a check for changes or variation in organic carbon concentration between Field sediments and those sediments that were actually incubated. As a quality assurance procedure (detailed below), we removed samples with the largest changes in organic carbon concentration between Field and Incubation sediments.

Field and Incubation sediments were extracted with milli-Q water, and the resulting supernatant from sediment extractions was filtered through a 0.22 μm Sterivex filter (EMD Millipore). Non-purgeable organic carbon (NPOC) was determined on the supernatant by a Shimadzu combustion carbon analyzer TOC-L CSH/CSN E100V with ASI-

L autosampler. We only included data from sites that had similar NPOC concentrations between the paired Field and
Incubation samples. Our rationale for this approach is based on the assumption that if NPOC is highly variable
across replicate sub-samples (i.e., across paired Field and Incubation samples), the associated sediments used for
respiration measurements may have been highly heterogeneous despite our efforts to homogenize sediments prior to
analyses. In turn, we assume that high heterogeneity may lead to unreliable estimates of NPOC, respiration, and OM
molecular richness for a given site. Focusing analyses on the subset of sites that had relatively good correspondence
in NPOC between Field and Incubation samples is, therefore, a conservative approach aimed at working with only
the most reliable data.

To subset the data, we calculated the ratio between Field and Incubation NPOC concentrations within each site. If
the ratio was less than 1, it was inverted so that all  ratios were greater than 1 because the important consideration
was the proportional difference between the Field and the Incubation NPOC concentrations. The same proportional
difference could lead to ratios below or above 1 depending on whether Field or Incubation NPOC was higher. For
our analysis we needed to know the proportional difference, not whether Field NPOC was higher or lower than
Incubation NPOC. In turn, we inverted the Field-to-Incubation NPOC ratio if it was below 1 so that all proportional
differences were more quantitatively comparable. We then regressed log-transformed Field NPOC vs. log-
transformed Incubation NPOC, and calculated the $R^2$ of the associated regression. Log-transformation was used due
to the presence of skewed NPOC distributions. Subsequently, we removed samples in order of their ratio, starting
with the largest ratio (i.e., the largest proportional difference between Field and Incubation NPOC). Higher $R^2$
values indicated a tighter relationship between Field and Incubation NPOC, and thus more reliable data. We
repeated these steps for all the samples in the Field-Incubation dataset (n = 228). We then plotted the $R^2$ vs. the
number of samples removed and selected a threshold for the number of samples to remove (Fig. S1). The resulting
curve showed that $R^2$ increased as a function of points removed until it leveled off. This nonlinear saturating
relationship was well-described by a Michaelis-Menten function (Michaelis and Menten, 1913; Johnson and Goody,
2011). In this function, the half saturation constant indicates the resource availability at which half of the maximum
intake is reached (Mulder and Hendriks, 2014). We used the half saturation constant, estimated from fitting the
function to the data in Fig. S1, in a conceptually analogous way. That is, the half saturation constant indicated the
number of samples that would need to be removed to gain half of the maximum potential increase in fit between
Field and Incubation NPOC. This resulted in removing 30 samples, leading to $R^2 = 0.74$ for the relationship between
Field and Incubation NPOC, which was half way between a minimum $R^2 = 0.47$ and maximum $R^2 = 1$. This
procedure was used to increase the reliability of the OM molecular richness estimates by removing samples that had
the greatest variability in NPOC, which could translate into variability in OM richness as there was a weak but
significant relationship between OM richness and NPOC ($R^2 = 0.20$, $p < 0.001$, Fig. S2).

Subsetting the data is a data quality control challenge and there are a variety of ways in which one could approach it.
In all quality control approaches there is a tradeoff between increasing confidence in data and removing so much
data that statistical analyses become impossible. We aimed to increase data confidence up to an inflection point
beyond which there appeared to be diminishing returns. Based on the functional form of the data, it appeared that a
Michaelis-Menten function fit the data very well. This functional form also has the useful feature of estimating the
half saturation constant, which we considered to be a practically useful inflection point.

*Fourier Transform Ion Cyclotron Resonance Mass Spectrometry (FTICR-MS)*
We used ultrahigh resolution Fourier transform ion cyclotron resonance mass spectrometry (FTICR-MS) to generate
mass spectra of sediment OM pools. Field sediment extracts were normalized to 1.5 mg C $L^{-1}$, acidified to pH 2 and
extracted with solid phase extraction (SPE) PPL cartridges following procedures described by (Dittmar et al., 2008).
Note that all samples were normalized to a consistent NPOC concentration prior to SPE and the same sample
volume was extracted with the same cartridges and resin mass. Since concentrations were normalized prior SPE, we
did not measure extraction efficiency post-extraction. While extraction efficiency will vary across samples, our
approach focuses on studying how many unique molecules were present in a sample rather than the relative
concentrations of individual molecules. We assume that variation in extraction efficiency is not systematically
linked to respiration rate to such a degree that the number of detected peaks becomes correlated with respiration.
Although we cannot definitively evaluate this assumption, we do not use information on relative peak intensities,
which should be influenced by SPE more than the number of peaks is influenced by SPE. It seems extremely
unlikely that the number of observed peaks would become systematically and spuriously linked to respiration due to
variation in extraction efficiency. In the worst case, biases would strengthen the statistical link between respiration
and the number of unique peaks, but we found this relationship to be very weak (see Results and Discussion).

FTICR-MS analyses were carried out at the Environmental Molecular Science Laboratory (EMSL) in Richland, WA
using a 12 Tesla (12T) Bruker SolariX FTICR mass spectrometer (Bruker, SolariX, Billerica, MA, USA) in negative
mode. The method used to assign molecular formulas to FTICR-MS spectra is described in Garayburu-Caruso et al.
(Garayburu-Caruso et al., 2020b). Briefly, Formularity (Tolić et al., 2017) was used to align mass lists generated
using Bruker DataAnalysis V4.2. Resulting reports were processed using ftmsRanalysis (Bramer et al., 2020). It is
important to note that FTICR-MS is a non-targeted approach to reliably identify molecular formulas of organic
molecules with masses, but it is not quantitative and does not provide information about the structure of the
molecular formulas identified. Our analyses on the Field FTICR-MS data only included samples that passed through
the subsetting process described above based on Field and Incubation NPOC. We calculated OM richness as the
total number of unique peaks present in one sample.

*Incubations and respiration rates*
Respiration rates were determined following methods described by Garayburu-Caruso et al. (2020a). Sieved
sediments were subsampled into 40 mL clear glass vials (I-Chem amber VOA glass vials) with a 0.5 cm diameter
factory calibrated oxygen sensor dot (Fibox 3; PreSens GmbH, Regensburg, Germany). Vials with sediments and
unfiltered water from each site were kept in the dark inside the environmental chamber at a 21°C until next day
incubations. Reactors consisted of 10 mL of sieved sediments and ~30-35 mL of aerated unfiltered water with no
headspace, shaken at 250 rpm for 2 hours. Dissolved oxygen (DO) was measured noninvasively every 15 min for
the first hour and every 30 min during the second hour using an oxygen optical meter (Fibox 3; PreSens GmbH,
Germany) to read the oxygen sensor dots on the vials. Respiration rates were calculated as the slope of the linear
regression between DO concentration and incubation time for each reactor and further normalized per gram of
sediment in each reactor. Normalized and not-normalized rates are reported in this manuscript.

*Statistical analysis*
All statistical analyses were completed using R (version 3.6.3)(R Core Team, 2021) with $p < 0.05$ as the significance
threshold. We used ordinary least squares regressions (function "lm") to evaluate relationships between respiration
rates and OM richness or NPOC. While not initially expected, we observed an apparent non-linear constraint-based
relationship between respiration rate and the inverse of NPOC. To evaluate the statistical significance of the
constraint boundary, we subdivided the 1/NPOC data into 10 even bins and found the maximum respiration rate in
each of those bins. We then fit a negative exponential function to the relationship between maximum respiration
rates and the 1/NPOC values associated with those maxima (i.e., we did not use the average 1/NPOC of each bin).
To evaluate the potential contribution of OM richness, we used the same approach to regress respiration rate against
the ratio of OM richness to NPOC concentration. Base functions in R and ggplot2 (Wickham, 2016) were used for
these analyses and associated plotting.

Scripts necessary to reproduce the primary results of this manuscript are available at
https://github.com/WHONDRS-Hub/Respiration_and_OM_Richness/ . Goldman et al. (2020) provides the raw,
unprocessed FTICR-MS data and respiration rate data. FTICR-MS data used in this manuscript were processed
following instructions provided in the Goldman et al. (2020) data package.

**3 Results and Discussion**
Both respiration rates and OM molecular richness varied significantly across samples, providing a useful dataset to
study the hypothesized negative relationship between these two variables. More specifically, the distribution of
aerobic respiration rates revealed a broad range of rates that were highly skewed for rates that were either not
normalized (Fig. 2A) or were normalized (Fig. 2B) per gram of sediment. These skewed distributions indicate the
potential for biogeochemical "hot spots" (McClain et al. 2003) or "control points" (Bernhardt et al. 2017) at the
continental scale. The distribution for OM molecular richness (i.e., number of identified organic molecules)
appeared to be multimodal, but dominated by one primary peak (Fig. 2C). OM molecular richness ranged from
~2000-5000 peaks, and we took advantage of this variation to evaluate the relationship between OM richness and
respiration rates.

We did not observe a clear negative relationship between sediment aerobic respiration rates and OM molecular
richness, which rejected the hypothesis that higher OM richness will suppress respiration (Fig. 3A,C).  The data in
Fig.3 suggest there may instead be a peak in maximum respiration rate near intermediate levels of OM molecular
richness. There may, therefore, be an optimal level of OM molecular richness that enables high respiration rates, but
does not guarantee elevated rates, leading to a unimodal constraint space. Regressions based on maximum
respiration rates across the OM richness axis were not significant, however (Fig. 3). These results further reject the
hypothesis of any direct relationship between respiration rate and OM richness.

While we did not observe a direct link between respiration and OM richness, extending the Lehmann et al. (2020)
hypothesis revealed a potential influence of OM richness, after controlling for water soluble organic carbon (OC)
concentration, measured as NPOC. That is, we posit that any connection between OM richness and respiration is
likely modified by the amount of OC. The magnitude of OM richness relative to the concentration of OC could,
therefore, provide a stronger constraint over respiration than OM richness alone. High ratios indicate high levels of
OM richness relative to the amount of OC, while low ratios indicate low levels of OM richness relative to the
amount of OC. In the context of the Lehmann et al. (2020) hypothesis, respiration would therefore be expected to
decrease with increasing richness-to-concentration ratios.

Consistent with this extended hypothesis, we find that maximum respiration rates decreased with increasing
richness-to-concentration ratios (Fig. 4). This suggests that at the continental scale OM molecular richness may
indirectly influence aerobic respiration rates. However, the influence of OM richness is likely to be relatively minor.
That is, maximum respiration rate was also well-explained simply to the inverse of OC concentration (Fig. 5). Note
that regression models applied to the whole datasets presented in Figure 4 and Figure 5, were relatively weak when
compared to models of the constraint boundaries ($R^2$ = 0.32-0.34 vs. $R^2$ = 0.89-0.97, Table S1). This further supports
our inference of respiration rates being constrained based on the richness-to-concentration ratio. Additionally, the
relationship between respiration rates and OC concentrations was relatively weak (Fig. S3). The statistical models
using the richness-to-concentration ratio are technically better models as they have higher $R^2$ values than models
using only the inverse of OC concentration (cf. Figs. 4, 5). We also note that both types of models are univariate, so
there is no penalty for multiple explanatory variables. The bulk of variation in maximum respiration rates (~90%) is,
however, explained simply by the inverse of OC concentration.

We infer that OC concentration could impose a primary constraint over maximum respiration rates, with OM
richness potentially contributing additional constraints. As such, any influences of OM richness over respiration
rates in hyporheic zone sediments are likely modulated by OC concentration. This is conceptually consistent with
observations in marine (Arrieta et al., 2015) and river corridor systems (Stegen et al., 2018). That is, when OM
molecular richness is high relative to OC concentration, the probability of a microbe repeatedly encountering the
same type of molecule is minimized. In that case, the costs of maintaining metabolic machinery to metabolize any
specific type of molecule may outweigh the energy gains (Arrieta et al., 2015). When costs outweigh gains,
respiration is expected to be minimized, which is consistent with the respiration constraint boundary to be lowest at
the highest richness-to-concentration ratios. In addition, the constraint boundary was non-linear, which is most likely
due to the fact that respiration rates cannot be below zero such that increasingly large richness-to-concentration
ratios cannot further suppress respiration.

The combined influences of OM richness and OC concentration are realized as a non-linear constraint space with the
vast majority of measured respiration rates falling well below the constraint boundary. This indicates that in most
cases, additional controls over respiration drive respiration below its potential maximum. Discerning these
additional controls is an important avenue of future work. For example, it would be useful to evaluate the degree to
which microbial community diversity, composition, biomass, and/or functional potential are related to deviations
from the constraint boundary. In addition, the FTICR-MS data used here provides presence-absence of organic
molecules, but not relative abundances of organic molecules. Accounting for among-molecule variation in
concentrations could provide insights into factors driving respiration below the constraint boundary. Any potential
biases introduced by solid phase extraction (see Methods) and other methodological details would, however, need to
be accounted for prior to including information on relative abundances. In addition, we removed data points that had
high measurement uncertainty (see Methods) as these could mask true relationships. This focused our analyses on
samples with relatively homogenous sediments. Locations with very heterogeneous sediments, even after sieving to
< 2mm, may be important to capture in future analyses. One approach to meet this challenge is analyzing a small
number of technical replicates (~3) for all locations, examine variation among them, and analyze numerous (>10)
additional technical replicates for locations with the most heterogeneous sediments. This would be an efficient way
to enable robust use of all sampled systems. Geography is another aspect that needs broader consideration in future
efforts. The current study was limited to the ConUS and within that domain there were some poorly sampled regions
(Fig. 1) due to logistical limitations. Improved and expanded geographic sampling may not change the constraint
boundary itself, but will likely help discern what drives variation below the boundary. More generally, there are
many potential influences spanning physical (e.g., sediment texture), chemical (e.g., mineralogy), and biological
(e.g., fungal-to-bacterial ratios) features that could modulate the location of any given sample or system relative to
the constraint boundary quantified here. Exploring system heterogeneity in context of these additional features will
be helpful to understand what drives samples/systems below the constraint boundary.

The outcomes of our study are useful for guiding models towards and away from features and processes that need to
be represented to enable predictions of river corridor biogeochemical function. For large scale models, it appears
there is a constraint envelope for hyporheic zone respiration rates that is related primarily to OC concentration and
even more strongly to OM richness relative to OC concentration. This constraint boundary emerged from sites
distributed across the ConUS (Fig. S4), indicating that it is transferable across a broad range of river corridor
systems. The richness-to-concentration ratio, therefore, offers a simple way to represent variation in maximum
respiration rates across river corridors. Furthermore, for sites without estimates of OM richness, our results suggest
that variation in maximum respiration rate could be reasonably estimated via OC concentration alone. The constraint
spaces observed here could also be included in models more mechanistically whereby they would emerge from the
representation of how microbial metabolism is influenced by both OM molecular richness and OC concentration.
The constraint spaces could, alternatively, be included more phenomenologically in models through probabilistic
sampling respiration rates within the constraint space. Regardless of how the constraint boundaries are represented
in models, they have important implications for fundamental and applied aspects of river corridors. Systems that are
close to the boundary will consume more oxygen within their sediments, potentially releasing more $CO_2$ (Saccardi
and Winnick, 2021) and having negative influences on fish embryos (Jensen et al., 2009), but positive influences on
contaminant transformations (Fischer et al., 2005; Lewandowski et al., 2019). Knowledge of what environmental
factors move systems closer to or further away from the constraints boundaries will, therefore, be helpful in making
decisions about how to manage river corridors.
There may be additional aspects of OM molecular richness and chemical diversity (e.g., thermodynamic properties,
elemental ratios) that influence respiration and other biogeochemical rates in hyporheic zone sediments (e.g.,
Garayburu-Caruso et al., 2020a; Song et al., 2020). There are rich opportunities for future studies to explore links
between respiration rates and a broad range of univariate and multivariate OM diversity measures quantified through
metrics such as Rao's entropy (Mentges et al., 2017) and dendrogram-based methods (Danczak et al., 2020, 2021;
Hu et al., 2022). There is a need to examine such possibilities at both local and global scales, and to extend our
river-focused analyses to additional ecosystem types. The methods, data, and metadata from this study are all
used/formatted consistently to enable expansion of this dataset to more rivers and/or other systems such as soils and
marine sediments. Continuing to use the approaches and formats established here will facilitate synthesis and, in
turn, knowledge of what patterns and processes are or are not transferable across ecosystems.
**4 Code availability:** Scripts necessary to reproduce the primary results of this manuscript are available at
https://github.com/WHONDRS-Hub/Respiration_and_OM_Richness/.
**5 Data availability:** Data were published previously in Goldman et al. (2020) in the ESS-DIVE repository and are
licensed for reuse under the Creative Commons Attribution 4.0 International License.
**6 Author contributions:** JCS and VAG-C conceptualized the study, VAG-C performed analyses with feedback
from JCS, RED processed some of the data, AEG managed the sampling campaign, and LR, JMT, and JW
processed samples. JCS and VAG-C drafted the initial manuscript and all authors contributed to revisions.
**7 Competing interests:** The authors declare that they have no conflict of interest.
**8 Acknowledgements**
We would like to thank the Worldwide Hydrobiogeochemistry Observation Network for Dynamic River Systems
(WHONDRS) consortium members for collecting the samples used in this study. WHONDRS and the associated
data and science could not exist without efforts from the consortium. For this study, WHONDRS was supported by
the River Corridor Science Focus Area (SFA) at the Pacific Northwest National Laboratory (PNNL). The SFA is
supported by the U.S. DOE, Office of Biological and Environmental Research (BER), Environmental System
Science (ESS) Program. Some of the data used for this study were generated at the U.S. Department of Energy
(DOE) Environmental Molecular Science Laboratory User Facility. PNNL is operated by Battelle Memorial
Institute for the U.S. DOE under Contract No. DE-AC05-76RL01830. We thank Casey McGrath for generating
Figure 1 and providing the associated R scripts.

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

**Figures**

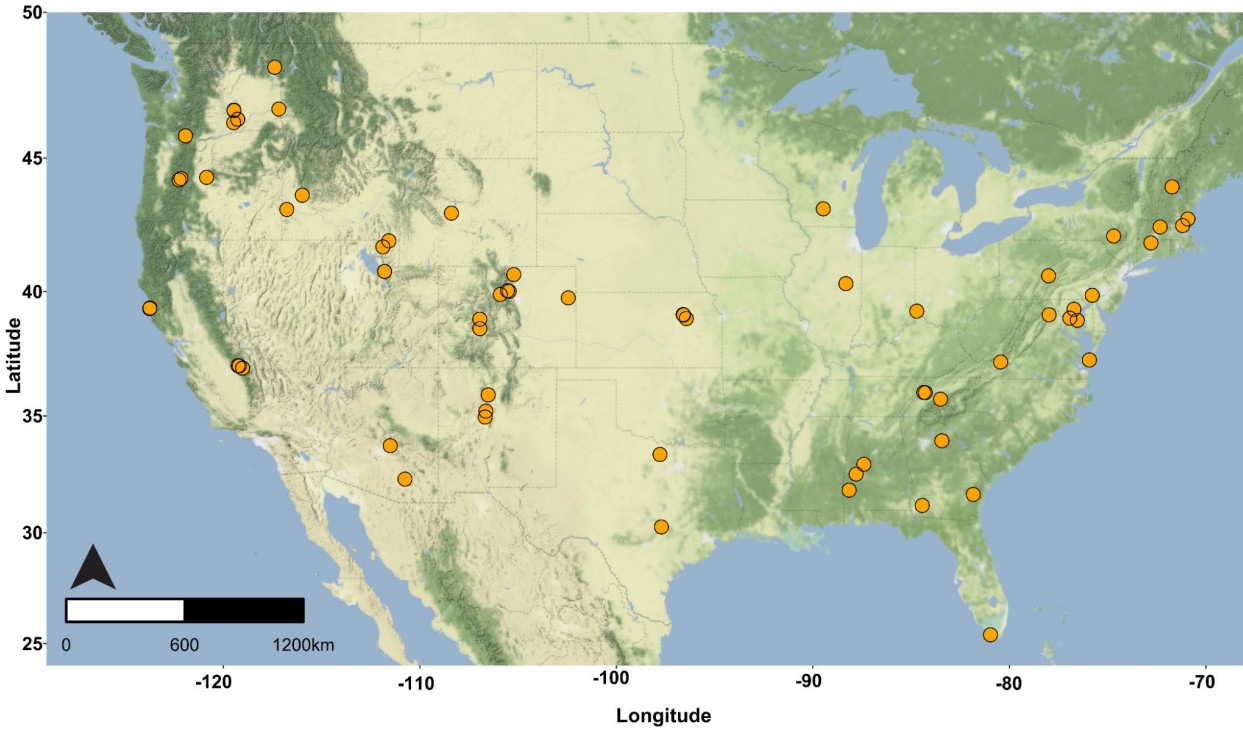

**Figure 1.** Spatial distribution of sampling locations. At each location three sediment samples were collected from locations
distributed along an upstream-downstream gradient within a single stream reach. The map was generated by Casey McGrath
using R via function get_stamenmap in package ggmap (Kahle et al., 2019). Map tiles by Stamen Design, under CC BY 3.0, and
the base map is copyrighted: ©OpenStreetMap contributors 2022. The base map is distributed under the Open Data Commons
Open Database 504 License (ODbL) v1.0.

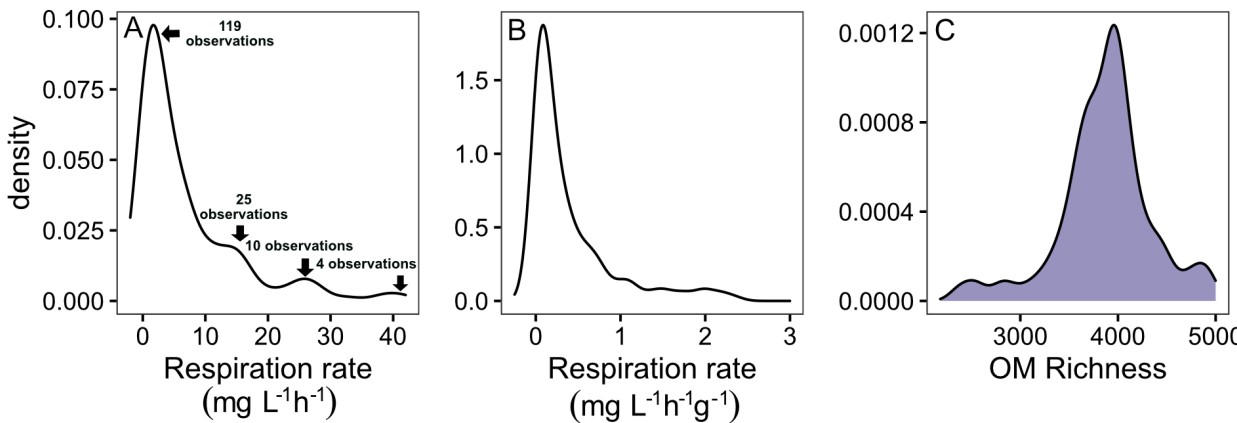

**Figure 2.** Density plots of aerobic respiration measured as oxygen consumption that was either not normalized relative to
sediment mass (A)  or normalized by sediment mass (B). Panel (C) is a density plot for the number of unique peaks identified in
sediment samples, which we refer to as OM richness.

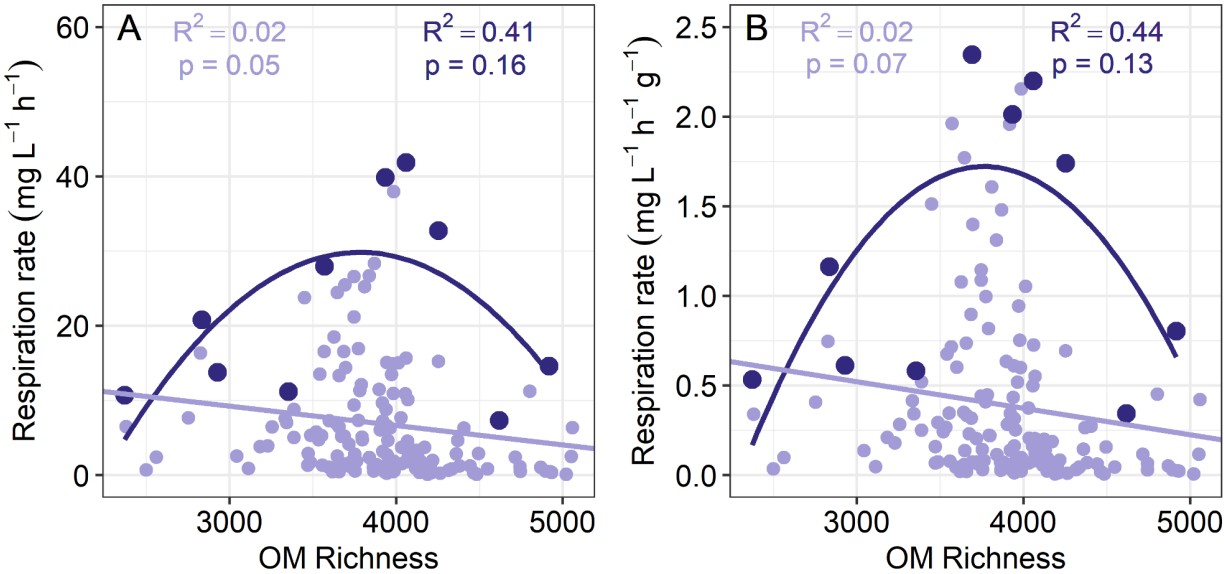


**Figure 3.** Sediment aerobic respiration as a function of OM richness. Respiration was measured as oxygen consumption and was
either not normalized (A) or normalized by sediment mass (B). Quadratic regression models based on maximum respiration rates
are shown in dark purple while linear regression models based on all respiration values are shown in light purple. Maximum
respiration rates were found by subdividing each horizontal axis into 10 even bins.  In all cases the models provided poor fits to
the data.

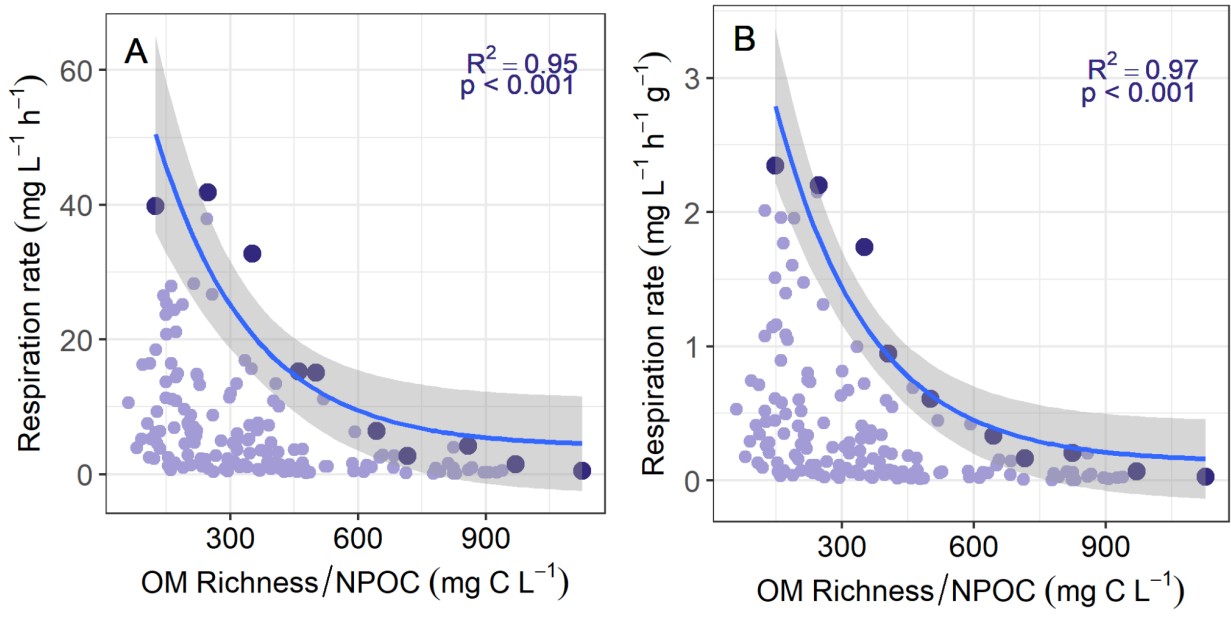


**Figure 4.** Maximum sediment respiration rate decreased with increasing values of ratio of OM molecular richness to non-
purgeable organic carbon concentration (NPOC). Panels A and B are for respiration that was either not normalized or normalized
by sediment mass, respectively. Maximum respiration rates (shown in the darker colors) were found by subdividing each
horizontal axis into 10 even bins. All other respiration rates and the corresponding richness-to-concentration ratios are shown in
lighter colors. Solid lines represent negative exponential models fit to the maximum respiration rates, with shaded areas
indicating 95% confidence intervals. Statistics for each model are provided on each panel. Statistics associated with exponential
regression models based on all respiration values (light purple) can be found in Table S1.

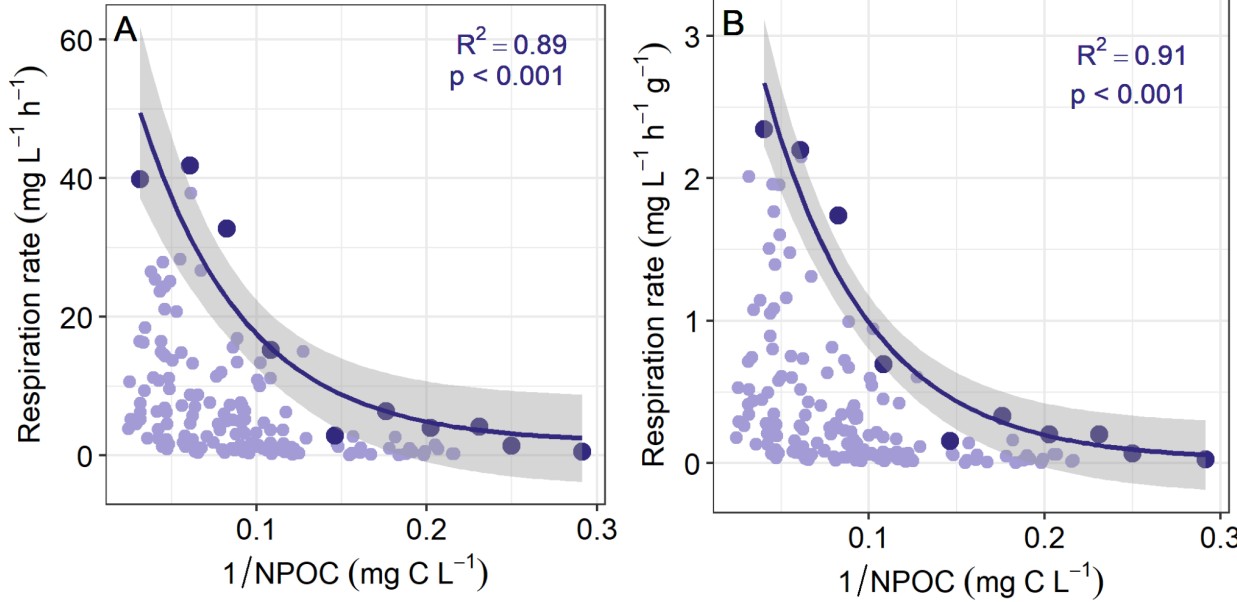


**Figure 5.** Maximum sediment respiration decreased with the inverse of non-purgeable organic carbon concentration (NPOC).
Panels A and B are for respiration that was either not normalized or normalized by sediment mass, respectively. Maximum
respiration rates (shown in the darker colors) were found by subdividing each horizontal axis into 10 even bins. All other
respiration rates and the corresponding 1/NPOC values are shown in lighter colors. Solid lines represent negative exponential
models fit to the maximum respiration rates, with shaded areas indicating 95% confidence intervals. Statistics for each model are
provided on each panel. Statistics associated with exponential regression models based on all respiration values (light purple) can
be found in Table S1.