# Peer review of "Hyporheic Zone Respiration is Jointly Constrained by Organic"

_EGUsphere, 2022_

## Author Response (AR1)

Dear Dr. Thurber,

Thank you for securing two high quality reviews. We followed the manuscript revision plan that you previously reviewed. Below please find our responses in bold text. In the material below we did our best to indicate the location of any edited text. We also uploaded a 'tracked changes' version of the manuscript.

We greatly appreciate your support and engagement and look forward to your further evaluation.

Sincerely,
James Stegen
Senior Research Scientist
Pacific Northwest National Laboratory
James.Stegen@pnnl.gov

############################

**Reviewer 1:**

Scientific significance:

The research concept is very clearly and simply stated in the Introduction. Hypotheses such as that framed and tested by the authors are challenging to undertake due to the difficulty in obtaining enough samples from a broadly distributed set of sites, using the same methods, in a timely fashion. Often these types of studies accrue data from multiple projects that were never intended to be considered collectively and so they may suffer from bias introduced because disparate research teams never coordinated and may have used different methods.  This research is an exception to that norm. This team has accomplished their research by using the WHONDRS program to contemporaneously collect a set of samples, using a common plan, to address their question.  The WHONDRS program sets a standard for how to carefully collect samples and corresponding reference data from a team of broadly distributed, motivated, self-selected collaborators and then to follow it up with detailed, systematic sample analysis.

**Thank you for sharing this encouraging perspective.**

Scientific Quality:

The use of the WHONDRS program's new and extensive database is notable and an exciting consideration of the data collected by the broad community of scientists who are in

support of WHONDRS.  I appreciate the use of standardized samples (Field vs. Incubation samples) as a way to account for possible heterogeneity in the samples and control for skewed results that might occur due to such heterogeneity. Does this approach mean that their conclusions are contingent on only acquiring samples from settings that are homogeneous?  In other words, do they only have confidence that their findings hold whenever samples are strictly homogeneous?  Is it possible that using this technique they've removed samples that are naturally heterogeneous and worth including despite this challenging feature?  This might be a consideration in the interpretation of the results.

**To address this interesting issue/question we added the following text in the Results and Discussion section of the paper (6th paragraph).**

**"In addition, we removed data points that had high measurement uncertainty (see Methods) as these could mask true relationships. This focused our analyses on samples with relatively homogenous sediments. Locations with very heterogeneous sediments, even after sieving to < 2mm, may be important to capture in future analyses. One approach to meet this challenge is analyzing a small number of technical replicates (~3) for all locations, examine variation among them, and analyze numerous (>10) additional technical replicates for locations with the most heterogeneous sediments. This would be an efficient way to enable robust use of all sampled systems. Geography is another aspect that needs broader consideration in future efforts. The current study was limited to the ConUS and within that domain there were some poorly sampled regions (Fig. 1) due to logistical limitations. Improved and expanded geographic sampling may not change the constraint boundary itself, but will likely be helpful to discern what drives variation below the boundary. More generally, there are many potential influences spanning physical (e.g., sediment texture), chemical (e.g., mineralogy), and biological (e.g., fungal-to-bacterial ratios) features that could modulate the location of any given sample or system relative to the constraint boundary quantified here. Exploring system heterogeneity in context of these additional features will be helpful to understand what drives samples/systems below the constraint boundary."**

I also wonder about the variation in extraction efficiency that the authors note in the Methods (pg 4).  What is the basis for the assumption that extraction efficiency is not systematically linked to respiration rate?  Is it possible that some compounds that are not easily extracted might also not be easily respired, i.e., that the "extractiveness" of a sample corresponds to its biological accessibility (in terms of respiration)?  The authors state that this assumption seems to be acceptable because it is extremely unlikely that extraction would be linked to respiration, but is there some evidence for this assumption?  Might it be possible that compounds exist that are both especially challenging to extract and challenging for microbes to respire?  Perhaps more background to support their assumption

would come from organic biogeochemistry studies that have considered the nature of recalcitrant compounds.

**We believe the reviewer is right in that there is technically a non-zero probability that there is some systematic link between extraction efficiency and respiration. While we believe such a systematic bias is very unlikely we nonetheless added some additional text as follows.**

**We added the following text to the 6th paragraph of the Methods:**

**"While extraction efficiency will vary across samples, our approach focuses on studying how many unique molecules were present in a sample rather than the relative concentrations of individual molecules. We assume that variation in extraction efficiency is not systematically linked to respiration rate to such a degree that the number of detected peaks becomes correlated with respiration. Although we cannot definitively evaluate this assumption, we do not use information on relative peak intensities, which should be influenced by SPE more than the number of peaks is influenced by SPE. It seems extremely unlikely that the number of observed peaks would become systematically and spuriously linked to respiration due to variation in extraction efficiency. In the worst case, biases would strengthen the statistical link between respiration and the number of unique peaks, but we found this relationship to be very weak (see Results and Discussion)."**

**We also added the following text to the 6th paragraph of the Results and Discussion:**

**"Any potential biases introduced by solid phase extraction (see Methods) and other methodological details would, however, need to be accounted for prior to including information on relative abundances."**

This is a relatively high-level view of the processes associated with the respiration of organic carbon in the hyporheic zone. Regarding a more detailed inspection of the data, a couple things come to mind and these might be helpful to point out in the discussion. Recognizing that the study was focused on trends that might be evident at the continental scale and accordingly required a collection of samples from a geographically vast area, it seems that there are some sample types that were not considered in the broad sampling effort. Presumably, this is because collaborators could not be recruited from these areas to collect samples. This might mean that certain watershed types as defined by regional climatic conditions, vegetation type, edaphic quality, regolith, underlying geology, stream gradient, etc. would have been under-represented in the dataset. From the map, examples of missing areas seem to be rivers located in the upper Plains, the Basin and Range Province, and on the Pacific coast. I cannot say that any such missing watersheds or river

systems are critical to their story, but omission of these regions in this study suggest that they should be included in future studies.

**We agree that there are some missing spatial and environmental domains. For the current manuscript we now point out this caveat in the 6th paragraph in the Results and Discussion section. We also note that we are currently running a crowdsourced sampling campaign to help fill these gaps. The added text is:**

**"Geography is another aspect that needs broader consideration in future efforts. The current study was limited to the ConUS and within that domain there were some poorly sampled regions (Fig. 1) due to logistical limitations. Improved and expanded geographic sampling may not change the constraint boundary itself, but will likely be helpful to discern what drives variation below the boundary."**

A tangential question: Are there systems other than marine and river corridors (as referenced at the bottom of pg 6) for which OM diversity and microbial respiration may have been considered? Could soils and marine sediments be added to their list and considered in this regard and if so, then could they also be interesting reference points, or distinctive contrasts for this study performed on samples from hyporheic zones? Conceivably, because of the way the WHONDRS work is conducted, this paper may be something of a landmark in having studied such a broad sweep of sample locations and might be used for future comparisons of microbially dominated ecosystems.

**At the moment we are unaware of other ecosystem types that have been sampled and analyzed using the same methods used for this study. Consistency in methodology will be important for quantitative comparisons. However, we very much appreciate the idea of using the current dataset as one that can be added to for quantitative comparisons across diverse ecosystems. The dataset is open access and well-structured to enable reuse (i.e., it is as FAIR as we could make it) and expansion.**

**To include the reviewer's (very nice) idea, we added the following text to the last paragraph of the Results and Discussion section:**

**"There is a need to examine such possibilities at both local and global scales, and to extend our river-focused analyses to additional ecosystem types. The methods, data, and metadata from this study are all used/formatted consistently to enable expansion of this dataset to more rivers and/or other systems such as soils and marine sediments. Continuing to use the approaches and formats established here will facilitate synthesis and, in turn, knowledge of what patterns and processes are or are not transferable across ecosystems."**

The Introduction includes reference to the importance of studying respiration in hyporheic zones. I think the Discussion could be improved by returning to this point and considering the how the findings may impact critical processes occurring in these riverine settings (i.e., where development or survival of larval/juvenile stages of fish species or aquatic invertebrates is fostered, where contaminant degradation occurs, where cold water refugia become important as rivers warm). The authors consider this at the end (lines 240-247) ; however, I think something more about the implications of the paper findings would be helpful to include here and would underline the importance of the work to readers.

**Thank you for another nice suggestion. We included additional material in the 7th paragraph of the Results and Discussion focused more on the implications for a broad audience. The added text is:**

**"Regardless of how the constraint boundaries are represented in models, they have important implications for fundamental and applied aspects of river corridors. Systems that are close to the boundary will consume more oxygen within their sediments, potentially releasing more CO2 (Saccardi and Winnick, 2021) and having negative influences on fish embryos (Jensen et al., 2009), but positive influences on contaminant transformations (Fischer et al., 2005; Lewandowski et al., 2019). Knowledge of what environmental factors move systems closer to or further away from the constraints boundaries will, therefore, be helpful in making decisions about how to manage river corridors."**

Presentation Quality:

This paper is a concise, straightforward, and articulate test of a well-stated hypothesis using a large and unique dataset. As a high-level view of the observed relationships between microbial respiration and OM diversity or OC concentration the paper succeeds in presenting the information. The figures are all appropriate for explaining their observations.

**Thank you for the encouraging thoughts.**

I suggest that the authors acknowledge the time and care taken by numerous scientists who sampled rivers and then contributed the hundreds of samples that were subsequently analyzed by the WHONDRS program. I'm certain that the original WHONDRS paper does so; however, it seems appropriate to have such a statement in all of the papers using data from this program.

**Another great suggestion, thank you. We now acknowledge the WHONDRS consortium members within the Acknowledgements as suggested.**

###################

**Reviewer 2:**

Does the paper address relevant scientific questions within the scope of BG?

Yes – it reaffirms the importance of organic carbon (OC) concentration as a major control on hyporheic zone respiration and offers suggestions for further relevant research and suggests a possible role of Organic matter molecular richness on hyporheic zone respiration.

**Thank you for the encouraging response.**

Does the paper present novel concepts, ideas, tools, or data?

Mostly – more effort could have been made to fully analyse the data available in order to achieve the stated goals.

**Please see our responses below associated with related reviewer feedback.**

Are substantial conclusions reached?

Yes

**Thank you for the encouraging response.**

Are the scientific methods and assumptions valid and clearly outlined?

Yes, but effort can be made to further validate assumptions/methods

**Please see our responses below associated with related reviewer feedback.**

Are the results sufficient to support the interpretations and conclusions?

Yes, the authors clearly state the importance of organic carbon (OC) concentration (as found by other authors) as a major control on hyporheic zone respiration, and that OM richness may have an influence on hyporheic zone respiration

**Thank you for the encouraging response.**

Is the description of experiments and calculations sufficiently complete and precise to allow their reproduction by fellow scientists (traceability of results)?

Yes

**Thank you for the encouraging response.**

Do the authors give proper credit to related work and clearly indicate their own new/original contribution?

Yes, but more credit should be given to the contributors of the WHONDRS dataset as the work presented here is an analysis of data collected and analysed by a wide range of contributors

**Thank you for this very important suggestion. We now acknowledge the WHONDRS consortium members within the Acknowledgements as suggested.**

Does the title clearly reflect the contents of the paper?

The title overstates the findings with respect to the authors findings in terms of molecular richness- The authors themselves state in the Abstract "we found that organic carbon (OC) concentration imposes a primary constraint over hyporheic zone respiration, with additional potential influences of OM richness." May I suggest to avoid any possible misunderstanding, that the title be adjusted to reflect the quoted text. Also further discussed below with respect to the use of respiration maxima.

**We changed the title to: "Maximum Respiration Rates in Hyporheic Zone Sediments are Primarily Constrained by Organic Carbon Concentration and Secondarily by Organic Matter Chemistry"**

**This is meant to address Reviewer 2's comment here and their comment below related to focusing on the maximum respiration rates. It also addresses Reviewer 1's comment about the potential bias introduced by variation in extraction efficiency, whereby the most robust inference is that there is something related to organic matter chemistry that has a secondary relationship with respiration rates (i.e., there is some chance it's not molecular richness, but rather something deeper about the molecular properties of the organic molecules; see our responses to Reviewer 1). The revised title uses the more general language of 'organic matter chemistry' to allow for this possibility.**

Does the abstract provide a concise and complete summary?

Yes

**Thank you for the encouraging response.**

Is the overall presentation well structured and clear?

Yes

**Thank you for the encouraging response.**

Is the language fluent and precise?

Yes

**Thank you for the encouraging response.**

Are mathematical formulae, symbols, abbreviations, and units correctly defined and used?

Yes

**Thank you for the encouraging response.**

Should any parts of the paper (text, formulae, figures, tables) be clarified, reduced, combined, or eliminated?

Yes, see detailed comments

**Please see our responses below associated with related reviewer feedback.**

Are the number and quality of references appropriate?

Yes

**Thank you for the encouraging response.**

Is the amount and quality of supplementary material appropriate?

Yes, but more effort could be made to potentially analyse the data in more detail, possibly leading to further significant scientific findings and conclusions.

**Please see our responses below associated with related reviewer feedback.**

Detailed comments

The authors aim to test and advance a proposed hypothesis from Lehmann et al. (2020) and seek to test this hypothesis of the presence of a negative relationship between respiration rates and OM molecular richness in the hyporheic zone on a continental scale using data collected from the WHONDRS consortium. The hyporheic zone is chosen due to

its higher levels of hydrologic connectivity which may diminish influences of spatial isolation such as an OM stabilization mechanism. The authors research rejects the hypothesis of any direct relationship between respiration rate and OM richness, both using the full dataset of sample respiration rates and maximum respiration rates across the OM richness. The authors confirm previous findings that OC concentration could impose a primary constraint over maximum respiration rates, with OM richness acting as a potential additional (but less important) constraint. The authors use maximum respiration rates to show that the combined influences of OM richness and OC concentration are realized as a non-linear constraint space, with the vast majority of measured respiration rates falling well below the constraint boundary. They further suggest research into additional factors which act as controls over respiration, which drive respiration below its potential maximum. The significant relationship between OM richness / NPOC and respiration rate is only valid for the respiration maxima and not for all the data collected, this seriously limits this continental scale study to a very small dataset. I would be interested to know the model results for the entire dataset of Respiration rate vs OM richness / NPOC (similar to the other models done) shown in Figure 4. I believe the title again does not reflect this important detail of the study findings and could lead to misunderstandings. Maybe a title along the lines of "Maximum respiration rates in the subsurface of rivers is predominantly constrained by organic carbon concentration, modulated by molecular richness" may be more representative.

**There are two primary points here, related to data analysis and the title.**

**For the model with the entire dataset of respiration rate vs. richness/NPOC, we now include those regression statistics in the supplemental material (Table S1). This includes models applied to the whole datasets presented in Figures 4 and 5. Given the non-linear nature of the relationship we fit and report results for negative exponential models. This is the same functional form fit to the constraint boundary. Direct quantitative comparison in terms of model fits (i.e. $R^2$ values) shows that while the fits to the whole data sets provide significant relationships (p-value << 0.001) the strength of the regressions are relatively weak ($R^2$ = 0.32-0.34 vs. $R^2$ = 0.89-0.97). We added the following text to the 4th paragraph in the Results and Discussion section:**

**"Note that regression models applied to the whole datasets presented in Figure 4 and Figure 5, were relatively weak when compared to models of the constraint boundaries ($R^2$ = 0.32-0.34 vs. $R^2$ = 0.89-0.97, Table S1). This further supports our inference of respiration rates being constrained based on the richness-to-concentration ratio."**

**The title has been revised as discussed above.**

L9-10: I would be cautious with the phrasing here to avoid a misinterpretation – What is the definition of the hyporheic zone referred to ? To my knowledge most definitions, including

those of authors cited in the current manuscript (eg. Krause et al. 2011) define the hyporheic zone as a zone of mixing of shallow groundwater and surface water. Not all sections of the river bed subsurface exhibit surface and groundwater mixing.

**We now more clearly define our meaning of 'hyporheic zone' as definitions vary across researchers.**

**The following text was added to the 1st paragraph of the Introduction section:**

**"Here we define the hyporheic zone as shallow subsurface sediments through which surface water enters, moves through and at some point returns to the main channel."**

L15-17 / 25-26:  Since the hyporheic zone is specifically mentioned, is the data used from WHONDRS exclusively from the hyporheic zone (HZ)?

**We use the definition of the hyporheic zone as those sediments through which surface water enters and at some point returns to the surface water channel. Collections of sediments were restricted to shallow (~1-3 cm depth) fine-grained sediments. As such, we make the assumption that surface water moves through those sediments and returns at some point to the water channel. In turn, we assume that all samples are reasonably conceptualized as hyporheic zone sediments. We now include a more detailed description of our definition, assumptions, and sampling methods in the revised manuscript.**

**The following text was added to the 1st paragraph of the Introduction section:**

**"Here we define the hyporheic zone as shallow subsurface sediments through which surface water enters, moves through and at some point returns to the main channel."**

**The following text was added to the 1st paragraph of the Methods section:**

**"We conceptualize these sediments as part of the hyporheic zone as we make the assumption that the supply and exchange of nutrients and OM from the stream influences the biogeochemical processes experienced by the sediments. In turn, we assume that all samples came from shallow (~1-3 cm depth) hyporheic zone sediments through which surface water enters, moves through, and at some point returns to the main channel."**

L24-25: What are the potential "other variables" that the results indicate are secondary influences on Hyporheic zone respiration (other than OM concentration) ? Could the authors

hypothesise based on literature which exists on the topic? Maybe lability, presence/ density of double/triple bonds, ring structures ?

**This is a very interesting and important direction to be heading. We feel there are a broad range of possible mechanisms and we now very briefly point out some possibilities here in the Abstract.**

**The last sentence of the Abstract now reads: "An important focus of future research will identify physical (e.g., sediment grain size), chemical (e.g., nutrient concentrations), and/or biological (e.g., microbial biomass) factors that suppress hyporheic zone respiration below the constraint boundaries observed here."**

L31-33: I would stress here not only contaminant removal, but more relevant to the paper, increased $CO_2$ evasion (respiration) and DOM alteration within the HZ. Several papers exist on the topic eg. Nature Comms. and Scientific Reports

**We edited the Introduction section to more directly link the hyporheic zone to CO2 evasion.**

**The first few sentences of the Introduction now read:**

**"River corridors are key components of the Earth system that connect terrestrial landscapes to the ocean through the transport and transformation of organic matter (OM) and nutrients (Harvey and Gooseff, 2015; Schlünz and Schneider, 2000; Schlesinger and Melack, 1981). In addition, river corridors have strong connections to the atmosphere in terms of significant emissions of greenhouse gasses such as $CO_2$, contributing ~3.9 Pg $CO_2$-C $yr^{-1}$ to the atmosphere (Raymond et al., 2013; Drake et al., 2017). Within river corridors the hyporheic zone (Orghidan, 2010) can have a dominant influence over net metabolism and biogeochemical transformations (Boulton et al., 1998; Naegeli and Uehlinger, 1997; Krause et al., 2011)"**

L41-46: I would argue that the classification of the molecular diversity in terms of structural complexity (eg. presence and number of ring structures, C:H, C:O ratios, N containing molecular formulae potentially indicating proteins, etc) and not simply number of unique organic molecules (after all the authors present FTICR-MS data) is also important for this. I would be interested which effect the different fractions of DOM molecules have on respiration. Have the authors explored DOM diversity in the level? I think it would be very interesting to identify groups of molecules that lead to higher respiration rates versus other groups.

**This is a very interesting direction, though going down this path opens up a huge variety of analyses (e.g., >10 mean properties, Rao's functional diversity for each of >10 properties, and up to three dendrogram-based methods integrating across properties). Each of those 25-30 analyses will need to be modeled against respiration rates in terms of whole-dataset and maximum values in both univariate and multivariate regressions. In total that will lead to ~100 additional analyses, with associated figures and statistical models. This will greatly expand the number of required figures and length of the Results and Discussion, leading to a very different paper. One of the strengths, in our opinion, of the current paper is that it is very tightly focused with a clear message. Our preference is to point to the need/opportunity for these additional analyses in the Discussion of the manuscript. The editor has indicated their support for this approach, and we added the following text to the last paragraph of the Results and Discussion section:**

**"There are rich opportunities for future studies to explore links between respiration rates and a broad range of univariate and multivariate OM diversity measures quantified through metrics such as Rao's entropy (Mentges et al., 2017) and dendrogram-based methods (Danczak et al., 2020, 2021; Hu et al., 2022). There is a need to examine such possibilities at both local and global scales, and to extend our river-focused analyses to additional ecosystem types."**

L50-56: I am not convinced that all the data used from WHONDRS is actually from the hyporheic zone, can you confirm that it is ?

**Please also see our responses above. In short, sediments were collected from ~1-3 cm depth, relative to the riverbed surface, and we assume that surface water enters and flows through these shallow sediments and at some point returns back to the surface channel. Per the definition we will include in the manuscript, we consider this to be hyporheic exchange such that we consider the sediments to be part of the hyporheic zone. We now include the definition along with associated assumptions and field methods, as indicated above.**

L100 – 101: This seems counter intuitive to me. You inverted ratios that were less than 1 ? Please explain further

**To clarify our reasoning we edited the first sentences of paragraph 4 of the Methods section to now read:**

**"To subset the data, we calculated the ratio between Field and Incubation NPOC concentrations within each site. If the ratio was less than 1, it was inverted so that all**

**ratios were greater than 1 because the important consideration was the proportional difference between the Field and the Incubation NPOC concentrations. The same proportional difference could lead to ratios below or above 1 depending on whether Field or Incubation NPOC was higher. For our analysis we needed to know the proportional difference, not whether Field NPOC was higher or lower than Incubation NPOC. In turn, we inverted the Field-to-Incubation NPOC ratio if it was below 1 so that all proportional differences were more quantitatively comparable."**

L 106-118: Is the use of a Michaelis-Menten function and the half saturation truly more justifiable than the use of a least squares approach with a pre-determined limit on the tolerated difference between the "replicate" Field and Incubation NPOC samples (maybe 20%) that would justify removal. Please explain.

**To clarify our reasoning, we added the following text as paragraph 5 of the Methods section:**

**"Subsetting the data is a data quality control challenge and there are a variety of ways in which one could approach it. In all quality control approaches there is a tradeoff between increasing confidence in data and removing so much data that statistical analyses become impossible. We aimed to increase data confidence up to an inflection point beyond which there appeared to be diminishing returns. Based on the functional form of the data, it appeared that a Michaelis-Menten function fit the data very well. This functional form also has the useful feature of estimating the half saturation constant, which we considered to be a practically useful inflection point."**

L126-130: Would FTICR-MS not yield information on molecular formulae, C:H, C:N, C:O ratios and thus indicate apparent lability? This may give further useful information.

**This is related to a comment above about adding additional evaluations of organic matter chemistry to the paper. Our preference is to keep the paper's analyses as they are and not expand into a large suite of additional analyses. FTICR-MS data are incredibly rich in terms of offering nearly limitless ways of using the data to study organic matter chemistry. As noted above, we feel a strength of our paper is that we have avoided the temptation to include a huge variety of exploratory analyses, and instead have focused on specific analyses tied to specific hypotheses. For specifics of text edits that point to future analyses, please see our response to a related comment above.**

L161-165: Just for clarity, was the maximum respiration rate in each bin plotted against the corresponding 1/NPOC value for that respiration rate or against an average of the bin ?

**We revised the Statistical analysis section with the Methods to reflect that we used the 1/NPOC that corresponded to the maximum respiration rate as the value along the x-axis. It now reads:**

**"We then fit a negative exponential function to the relationship between maximum respiration rates and the 1/NPOC values associated with those maxima (i.e., we did not use the average 1/NPOC of each bin)."**

L177-179: A skewed distribution is a possible indicator of another key controlling factor that was not taken into account by the model / study, correct?

**At this point in the paper we are describing the distribution of measured rates and are not developing statistical models to explain variation in the data. Biogeochemical hot spots are broadly acknowledged as being essential components of ecosystems and their presence will, by definition, lead to skewed distributions of biogeochemical rates. In turn, we interpret the observation of a skewed distribution as indicating that we sampled enough sites to capture biogeochemical hot spots across the contiguous U.S. We find this an encouraging outcome of the study. In addition, we were able to develop highly explanatory statistical models of the constraint space that includes both low rates and the hot spots. In turn, we feel that we have accounted for the necessary factors, given the goals of our study.**

L185-188: While the hypothesis sounds reasonable, I am not completely convinced by the data presented in the current graph. There are only three (out of ten) points making up the negative slope on the right of the graph showing a decrease in respiration rate with OM richness above 4000 unique peaks. The point representing the highest OM richness corresponds to almost double the respiration rate of the point representing the bin before it. Maybe using the maxima from 15 or 20 bins would make the relationship clearer ?

**In this section we argue that there is no evidence to support the hypothesis that higher DOM richness leads to lower respiration rates. We believe the reviewer is saying the same thing here (i.e., the data presented in Fig. 3 are not consistent with the hypothesis). In turn, we believe there are no modifications to be made to the associated text. If we misinterpreted the reviewer's comments, we would be happy to reevaluate.**

L201 – 209 : Given the authors analysis of the results, is the title of the paper truly justified ? Is it possibly a bit of an overstatement of the role of OM richness ? Should the title reflect more the statements in L 211 – 212?

**Please see above for discussion on our revisions to the title.**

Figure 1: It seems that the samples were biased toward rivers in lower altitudes and flatter terrain (possibly lower gradient rivers?), as well as away from the central section of the USA. Could this have excluded some important environments/factors that are important for a "continental scale" model ? Also, what does the map look like showing the spatial distribution of final data point locations that were analysed for the model ?

**It is an important caveat for all observational studies that all outcomes can be made only with respect to the sampling locations that were used. As the reviewer notes, our sampling did miss some parts of the contiguous U.S., and in particular the upper midwest region. We did, however, sample across a broad range of environmental conditions such as stream order (1$^{st}$ to 8$^{th}$) and land cover compositions (e.g., forest cover ranging from 0-97 % and urban cover ranging from 0-28%). Given the breadth of sampled environments, we have confidence in our outcomes and inferences, but agree that it is appropriate to call out some caveats and limitations related to the distribution of sampling locations. Text summarizing these limitations will be added to the manuscript, likely in the Methods and in the Results and Discussion.**

**The following text was added to the 1st paragraph of the Methods section:**

**"Sampled locations spanned a broad range of environmental conditions; for example, stream order ranged from 1$^{st}$ to 8$^{th}$, land cover composition varied with upstream forest cover ranging from 0-97 % and urban cover ranging from 0-28%, and physical settings were from relatively steep headwater streams to large lowland rivers."**

**The following text was added to the 6th paragraph of the Results and Discussion:**

**"Geography is another aspect that needs broader consideration in future efforts. The current study was limited to the ConUS and within that domain there were some poorly sampled regions (Fig. 1) due to logistical limitations. Improved and expanded geographic sampling may not change the constraint boundary itself, but will likely be helpful to discern what drives variation below the boundary."**

**In addition, we included a map to show the spatial distribution of samples that defined the constraint space following the reviewer's suggestion. The map is Figure**

**S4 in the supplemental material of the revised manuscript. We added the following text to the 2nd to last paragraph of the Results and Discussion:**

**"This constraint boundary emerged from sites distributed across the ConUS (Fig. S4), indicating that it is transferable across a broad range of river corridor systems."**

Figure 4/5: The full dataset is shown here. Why wasn't a model for the full dataset calculated and results shown as comparison as done previously in Fig. 3 ?

**We have included the statistics within a supplemental table (Table S1) to summarize models applied to the whole datasets across Figures 4 and 5. Given the non-linear nature of the relationship we specifically fit and report on negative exponential models. This is the same functional form fit to the constraint boundary so also provides a useful and direct quantitative comparison in terms of model fits (i.e., we use $R^2$ values to compare models).**

**The following text was added to the 4th paragraph of the Results and Discussion:**

**"Note that regression models applied to the whole datasets presented in Figure 4 and Figure 5, were relatively weak when compared to models of the constraint boundaries ($R^2$ = 0.32-0.34 vs. $R^2$ = 0.89-0.97, Table S1). This further supports our inference of respiration rates being constrained based on the richness-to-concentration ratio."**

---

## Author Response (AR2)

Dear Dr. Thurber,

Thank you for your help in making our manuscript stronger.
We have uploaded all the final files.
We look forward to working with Biogeosciences again on future manuscripts.

Sincerely,
James Stegen
Senior Research Scientist
Pacific Northwest National Laboratory
James.Stegen@pnnl.gov